# MACHINE VS MACHINE: MINIMAX-OPTIMAL DEFENSE AGAINST ADVERSARIAL EXAMPLES

## ABSTRACT

Recently, researchers have discovered that the state-of-the-art object classifiers can be fooled easily by small perturbations in the input unnoticeable to human eyes. It is known that an attacker can generate strong adversarial examples if she knows the classifier parameters. Conversely, a defender can robustify the classifier by retraining if she has the adversarial examples. The cat-and-mouse game nature of attacks and defenses raises the question of the presence of equilibria in the dynamics. In this paper, we present a neural-network based attack class to approximate a larger but intractable class of attacks, and formulate the attacker-defender interaction as a zero-sum leader-follower game. We present sensitivity-penalized optimization algorithms to find minimax solutions, which are the best worst-case defenses against whitebox attacks. Advantages of the learning-based attacks and defenses compared to gradient-based attacks and defenses are demonstrated with MNIST and CIFAR-10.

## 1 INTRODUCTION

Recently, researchers have made an unsettling discovery that the state-of-the-art object classifiers can be fooled easily by small perturbations in the input unnoticeable to human eyes (Szegedy et al., 2013; Goodfellow et al., 2014b). Following studies tried to explain the cause of the seeming failure of deep learning toward such adversarial examples. The vulnerability was ascribed to linearity (Szegedy et al., 2013), low flexibility (Fawzi et al., 2015), or the flatness/curvedness of decision boundaries (Moosavi-Dezfooli et al., 2017), but a more complete picture is still under research. This is troublesome since such a vulnerability can be exploited in critical situations such as an autonomous car misreading traffic signs or a facial recognition system granting access to an impersonator without being noticed. Several methods of generating adversarial examples were proposed (Goodfellow et al., 2014b; Moosavi-Dezfooli et al., 2016; Carlini & Wagner, 2017), most of which use the knowledge of the classifier to craft examples. In response, a few defense methods were proposed: retraining target classifiers with adversarial examples called adversarial training (Szegedy et al., 2013; Goodfellow et al., 2014b); suppressing gradient by retraining with soft labels called defensive distillation (Papernot et al., 2016); hardening target classifiers by training with an ensemble of adversarial examples (Tramèr et al., 2017).

In this paper we focus on whitebox attacks, that is, the model and the parameters of the classifier are known to the attacker. This requires a more robust classifier or defense method than simply relying on the secrecy of the parameters as defense. When the classifier parameters are known to an attacker, existing attack methods are very successful at fooling the classifiers. Conversely, when the attack is known to the classifier, e.g., in the form of adversarial examples, one can weaken the attack by retraining the classifier with adversarial examples, called adversarial training. However, if we repeat adversarial sample generation and adversarial training back-to-back, it is observed that the current adversarially-trained classifier is no longer robust to previous attacks (see Sec. 3.1.) To find the classifier robust against the class of gradient-based attacks, we first propose a sensitivity-penalized optimization procedure. Experiments show that the classifier from the procedure is more robust than adversarially-trained classifiers against previous attacks, but it still remains vulnerable to some degrees. This raises the main question of the paper: *Can a classifier be robust to all types of attacks?* The answer seems to be negative in light of the strong adversarial examples that can be crafted by direct optimization procedures from Huang et al. (2015) or Carlini & Wagner (2017). Note that the class of optimization-based attack is very large, as there is no restriction on the

adversarial patterns that can be generated except for certain bounds such as $l_p$-norm bounds. The vastness of the optimization-based attack class is a hindrance to the study of the problem, as the defender cannot learn efficiently about the attack class from a finite number of samples. To study the problem analytically, we use a class of learning-based attack that can be generated by a class of neural networks. This class of attack can be considered an approximation of the class of optimization-based attacks, in that the search space of optimal perturbation is restricted to the parameter space of a neural network architecture, e.g., all perturbations that can be generated by fully-connected 3-layer ReLU networks. Similar to what we propose, others have recently considered training neural networks to generate adversarial examples (Nguyen & Sinha, 2017; Baluja & Fischer, 2017). While the proposed learning-based attack is weaker than the optimization-based attack, it can generate adversarial examples in test time with only single feedforward passes, which makes real-time attacks possible. We also show that the class of neural-network based attacks is quite different from the the class of gradient-based attacks (see Sec. 4.1.)

Using the learning-based attack class, we introduce a continuous game formulation for analyzing the dynamics of attack-defense. The game is played by an attacker and a defender/classifier [1], where the attacker tries to maximize the risk of the classification task by perturbing input samples under certain constraints such as $l_p$-norm bounds, and the defender/classifier tries to adjust its parameters to minimize the same risk given the perturbed inputs. It is important to note that for adversarial attack problems, the performance of an attack or a defense cannot be measured in isolation, but only in pairs of (attack, defense). This is because the effectiveness of an attack/defense depends on the defense/attack it is against. As a two-player game, there may not be a dominant defense that is no less robust than all other defenses against all attacks. However, there is a natural notion of the best defense or attack in the worst case. Suppose one player moves first by choosing her parameters and the other player responds with the knowledge of the first player's move. This is an example of a leader-follower game (Brückner & Scheffer, 2011) for which there are two well-known states, the minimax and the maximin solutions if it is a constant-sum game. To find those solutions empirically, we propose a new continuous optimization method using the sensitivity penalization term. We show that the minimax solution from the proposed method is indeed different from the solution from the conventional alternating descent/ascent and is also more robust. We also show that the strength/weakness of the minimax-trained classifier is different from that of adversarially-trained classifiers for gradient-based attacks. The contributions of this paper are summarized as follows.

- We provide a continuous game model to analyze adversarial example attacks and defenses, using the neural network-based attack class as a feasible approximation to a larger but intractable class of optimization-based attacks.

- We demonstrate the difficulty of defending against multiple attack types and present the minimax defense as the best worst-case defense methods.

- We propose a sensitivity-penalized optimization method (Alg. 1) to numerically find continuous minimax solutions, which is better than alternating descent/ascent. The proposed optimization method can also be used for other minimax problems beyond the adversarial example problem.

The proposed methods are demonstrated with the MNIST and the CIFAR-10 datasets. For readability, details about experimental settings and the results with CIFAR-10 are presented in the appendix.

## 2 RELATED WORK

Making a classifier robust to test-time adversarial attacks has been studied for linear (kernel) hyperplanes (Lanckriet et al., 2002), naive Bayes (Dalvi et al., 2004) and SVM (Globerson & Roweis, 2006), which also showed the game-theoretic nature of the robust classification problems. Since the recent discovery of adversarial examples for deep neural networks, several methods of generating adversarial samples were proposed (Szegedy et al., 2013; Goodfellow et al., 2014b; Huang et al., 2015; Moosavi-Dezfooli et al., 2016; Carlini & Wagner, 2017) as well as several methods of defense (Szegedy et al., 2013; Goodfellow et al., 2014b; Papernot et al., 2016; Tramèr et al., 2017). These papers considered static scenarios, where the attack/defense is constructed against a fixed opponent.

---

[1]The classifier and the defender are treated synonymous in this paper.

A few researchers have also proposed using a detector to detect and reject adversarial examples (Meng & Chen, 2017; Lu et al., 2017; Metzen et al., 2017). While we do not use detectors in this work, the minimax approach we proposed in the paper can be applied to train the detectors.

The idea of using neural networks to generate adversarial samples has appeared concurrently (Baluja & Fischer, 2017; Nguyen & Sinha, 2017). Similar to our paper, the two papers demonstrates that it is possible to generate strong adversarial samples by a learning approach. Baluja & Fischer (2017) explored different architectures for the "adversarial transformation networks" against several different classifiers. Nguyen & Sinha (2017) proposed "attack learning neural networks" to map clean samples to a region in the feature space where misclassification occurs and "defense learning neural networks" to map them back to the safe region. Instead of prepending the defense layers before the fixed classifier (Nguyen & Sinha, 2017), we retrain the whole classifier as a defense method. However, the key difference of our work to the two papers is that we consider the dynamics of a learning-based defense stacked with a learning-based attack, and the numerical computation of the optimal defense/attack by continuous optimization.

The alternating gradient-descent method for finding an equilibrium of a game has gained renewed interest since the introduction of Generative Adversarial Networks (GAN) (Goodfellow et al., 2014a). However, the instability of the alternating gradient-descent method has been known, and the "unrolling" method (Metz et al., 2016) was proposed to speed up the GAN training. The optimization algorithm proposed in the paper has a similarity with the unrolling method, but it is simpler (corresponding to a single-step unrolling) and involves a gradient-norm regularization which can be interpreted intuitively as sensitivity penalization (Gu & Rigazio, 2014; Lyu et al., 2015). Lastly, the framework of minimax risks was also studied in Hamm (2016) for the purpose of privacy preservation. We propose a different algorithm in this paper, but we also show that the attack on classification and the attack on privacy are the two sides of the same optimization problem with the opposite goals.

## 3 CAT-AND-MOUSE GAME

A classifier whose parameters are known to an attacker is easy to attack. Conversely, an attacker whose sample-generating method is known to a classifier is easy to defend from. In this section, we demonstrate the cat-and-mouse nature of the interaction, using adversarial training (Adv Train) as defense and the fast gradient sign method (FGSM) (Goodfellow et al., 2014b) and the iterative version (IFGSM) (Kurakin et al., 2016a) as attacks. We then show that the equilibrium, if it exists, can be found more efficiently by directly solving a sensitivity-penalized optimization problem.

### 3.1 A NAIVE APPROACH

Suppose $g$ is a classifier $g : \mathcal{X} \to \mathcal{Y}$ and $l(g(x), y)$ is a loss function. The FGSM attack generates a perturbed example $z(x)$ given the clean sample $x$ as follows:

$$z(x) = x + \eta \,\mathrm{sign}\,(\nabla_x l(g(x), y)). \tag{1}$$

The clean input images we use here are $l_\infty$-normalized, that is, all pixel values are in the range $[-1, 1]$. It was argued that the use of true label $y$ results in "label leaking" (Kurakin et al., 2016b), but we use will true labels in the paper for simplicity. For another attack example, the IFGSM attack iteratively refines an adversarial example by the following update

$$z_{i+1} = \mathrm{clip}_{x,\eta}(z_i + \eta \,\mathrm{sign}(\nabla_z l(g(z_i), y))), \tag{2}$$

where the clipping used in this paper is $\mathrm{clip}_{x,\eta}(x') \triangleq \min\{1,\ x + \eta,\ \max\{-1,\ x - \eta,\ x'\}\}$.

Existing attack methods such as FGSM and IFGSM are very effective at fooling the classifier. Table 1 shows that the two methods are able to perfectly fool a convolutional neural network trained with clean images from MNIST. (Details of the classifier architecture and the settings are in the appendix.)

On the other hand, these attacks, if known to the classifier, can be weakened by retraining the classifier with the original dataset augmented by adversarial examples with ground-truth labels, known as adversarial training. In this paper we use the 1:1 mixture of the clean and the adversarial samples for adversarial training. Table 2 shows the result of adversarial training for different attacks.

| Defense\Attack | No attack | FGSM | | | | IFGSM | | | |
|---|---|---|---|---|---|---|---|---|---|
| | | $\eta$=0.3 | $\eta$=0.4 | $\eta$=0.5 | $\eta$=0.6 | $\eta$=0.3 | $\eta$=0.4 | $\eta$=0.5 | $\eta$=0.6 |
| No defense | 0.006 | 1.000 | 1.000 | 1.000 | 1.000 | 1.000 | 1.000 | 1.000 | 1.000 |

Table 1: Test error rates of FGSM and IFGSM attacks on an undefended convolutional neural network for MNIST. These attacks can cause perfect misclassification for the given range of $\eta$.

The test error rates for adversarial test examples after training become below 1% indicating near-perfect avoidance. This is in stark contrast with the perfect misclassification of the undefended classifier in Table 1.

| Defense\Attack | No attack | FGSM | | | | IFGSM | | | |
|---|---|---|---|---|---|---|---|---|---|
| | | $\eta$=0.3 | $\eta$=0.4 | $\eta$=0.5 | $\eta$=0.6 | $\eta$=0.3 | $\eta$=0.4 | $\eta$=0.5 | $\eta$=0.6 |
| Adv train | n/a | 0.004 | 0.003 | 0.003 | 0.005 | 0.003 | 0.003 | 0.004 | 0.010 |

Table 2: Error rates of FGSM and IFGSM attacks on adversarially-trained classifiers for MNIST. This defense can avert the attacks and achieve the error rates of the no-attack case.

A question arises as to what would happen if the procedure of 1) adversarial sample generation using the current classifier, and 2) retraining classifier using the current adversarial examples is repeated for many rounds. The answer to this cat-and-mouse game is easy to experiment although time-consuming. Let's denote the attack on the original classifier as FGSM1, and the corresponding retrained classifier as Adv FGSM1. Repeating the procedure above generates the sequence of models FGSM1 → Adv FGSM1 → FGSM2 → Adv FGSM2, etc. Fig. 1 shows one such trial with 80 + 80 rounds of the procedure. Initially, the attacker achieves near-perfect attacks (i.e., error rate $\simeq 1$), and the defender achieves near-perfect defense (i.e., error rate $\simeq 0$). As the iteration increases, the attacker becomes weaker with error rate $\simeq 0.5$, but the defense is still very successful, and the rate seems to oscillate persistently. While we can run more iterations to see if it converges, this is not a very principled nor efficient approach to find an equilibrium, if it exists.

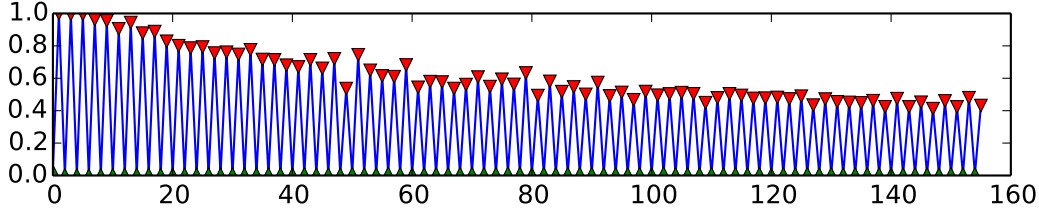

Figure 1: A cat-and-mouse game of FGSM attacks and adversarial training for MNIST. The upper red points are the error rates after adversarial training, and the lower green points are the error rates after FGSM attack ($\eta = 0.3$). After 160 iterations, the error rate is still oscillating between 0 and 0.5.

## 3.2 GRADIENT-BASED ATTACKS AND SENSITIVITY PENALTY

We can perform the cat-and-mouse simulation more efficiently by an optimization approach. Instead of training the classifier fully with adversarial examples and then regenerating adversarial examples, suppose we only update the classifier with a single gradient-descent step then regenerate adversarial examples. To emphasize the parameters $u$ of the classifier/defender $g(x; u)$, let's rewrite the empirical risk of classifying the perturbed data as

$$f(u, Z) \triangleq \frac{1}{N} \sum_{i=1}^{N} l(g(z(x_i); u), y_i), \tag{3}$$

where $z(x)$ denote an FGSM-like attack based on the loss gradient

$$z(x) \leftarrow x + \eta \, \nabla_z l(g(z(x); u), y), \tag{4}$$

and $Z = (z_1, ..., z_N) \triangleq (z(x_1), ..., z(x_N))$ is the sequence of perturbed examples. In expectation of the attack, the defender should choose $u$ to minimize $f(u, Z(u))$ where the dependence of the attack on the classifier $u$ is expressed explicitly. If we minimize $f$ using gradient descent

$$u \leftarrow u - \lambda \frac{df(u, Z)}{du}, \tag{5}$$

then from the chain rule, the total derivative $\frac{df}{du}$ is

$$\frac{df}{du} = \frac{\partial f}{\partial u} + \frac{\partial Z}{\partial u} \frac{\partial f}{\partial Z} = \frac{\partial f}{\partial u} + \sum_i \frac{\partial z_i}{\partial u} \frac{\partial f}{\partial z_i} = \frac{\partial f}{\partial u} + \frac{\eta}{N} \sum_i \frac{\partial^2 l}{\partial z_i \partial u} \frac{\partial l}{\partial z_i} \tag{6}$$

from (3) and (4).

Interestingly, this total derivative (6) at the current state coincides with the gradient $\nabla_u$ of the following cost

$$f_{\text{sens}}(u) \triangleq f(u, Z) + \frac{\gamma}{2} \left\| \frac{\partial f(u, Z)}{\partial Z} \right\|^2 = f(u, Z) + \frac{\eta}{2N} \sum_{i=1}^N \left\| \frac{\partial l(g(z_i; u), y_i)}{\partial z_i} \right\|^2 \tag{7}$$

where $\gamma = \eta N$. There are two implications. Interpretation-wise, this cost function is the sum of the original risk $f$ and the 'sensitivity' term $\|\partial f/\partial Z\|^2$ which penalizes abrupt changes of the risk w.r.t. the input. Therefore, $u$ is chosen at each iteration to not only decrease the risk but also to make the classifier insensitive to input perturbation so that the attacker cannot take advantage of large gradients. The idea of minimizing the sensitivity to input is a familiar approach in robustifying classifiers (Gu & Rigazio, 2014; Lyu et al., 2015). Secondly, the new formulation can be implemented easily. The gradient descent update using the seemingly complicated gradient (6) can be replaced by the gradient descent update of (7). The capability of automatic differentiation (Rall, 1981) in modern machine learning libraries can be used to compute the gradient of (7) efficiently. Using this direct approach, we can find the defense parameters $u$ which will be robust to gradient-based attacks. Fig. 2 shows the decrease of test error during training using the this gradient descent approach for MNIST. It only takes a very small fraction of time to reach the final states of the Fig. 2 compared to that of Fig. 1.

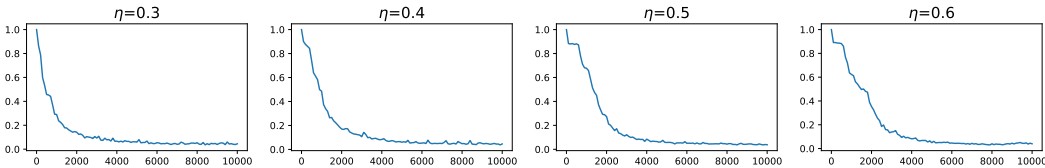

Figure 2: Convergence of test error rates for sensitivity-penalized optimization (7) with MNIST.

There is also an important difference between the solution of the cat-and-mouse game and the minimizer of (7). Table 3 shows that the adversarially trained classifier (Adv FGSM1) is robust to both clean data and FGSM1 attack, but is susceptible to FGSM2 attack, displaying the cat-and-mouse nature. The same holds for Adv FGSM2, Adv FGSM3, etc. After 80 rounds of the cat-and-mouse procedure, the classifier Adv FGSM80 becomes robust to FGSM80 as well as moderately robust to other attacks including FGSM81 (=FGSM-curr). However, the classifier Sens FGSM from direct minimization of (7) is even more robust toward FGSM-curr than Adv FGSM80 and is overall the best. To see the advantage of the sensitivity term in (7), we also performed the minimization of (7) without the sensitivity term under the same conditions as Sens FGSM. This optimization method is similar to the method proposed in Huang et al. (2015), referred to as Learning with Adversaries (LWA FGSM). In the table, one can see that Sens FGSM is also better than LWA FGSM overall, although the difference is small.

Note that Sens FGSM is better than other adversarially-trained classifiers, it too is still vulnerable to attacks such as FGSM80. This vulnerability raises the question if it is possible to make a classifier robust to any type of attacks, or more practically, robust to at least a large class of attacks. We discuss this issue in the next section.

| | Defense\Attack | No attack | FGSM | | | | FGSM-curr |
|---|---|---|---|---|---|---|---|
| | | | FGSM1 | FGSM2 | $\cdots$ | FGSM80 | |
| $\eta$=0.3 | No defense | 0.026 | 1.000 | 0.881 | $\cdots$ | 0.355 | 1.000 |
| | Adv FGSM1 | 0.012 | 0.004 | 0.995 | $\cdots$ | 0.499 | 0.995 |
| | Adv FGSM2 | 0.012 | 0.999 | 0.002 | $\cdots$ | 0.505 | 0.995 |
| | Adv FGSM80 | 0.009 | 0.335 | 0.273 | $\cdots$ | 0.009 | 0.442 |
| | LWA FGSM | 0.008 | 0.121 | 0.188 | $\cdots$ | 0.210 | 0.048 |
| | Sens FGSM | 0.009 | 0.104 | 0.176 | $\cdots$ | 0.194 | 0.048 |
| $\eta$=0.4 | No defense | 0.026 | 1.000 | 0.944 | $\cdots$ | 0.528 | 1.000 |
| | Adv FGSM1 | 0.013 | 0.003 | 0.984 | $\cdots$ | 0.589 | 0.984 |
| | Adv FGSM2 | 0.017 | 0.999 | 0.005 | $\cdots$ | 0.549 | 0.999 |
| | Adv FGSM80 | 0.009 | 0.509 | 0.525 | $\cdots$ | 0.024 | 0.131 |
| | LWA FGSM | 0.009 | 0.204 | 0.284 | $\cdots$ | 0.336 | 0.043 |
| | Sens FGSM | 0.009 | 0.128 | 0.234 | $\cdots$ | 0.296 | 0.038 |
| $\eta$=0.5 | No defense | 0.026 | 1.000 | 0.931 | $\cdots$ | 0.662 | 1.000 |
| | Adv FGSM1 | 0.010 | 0.002 | 0.970 | $\cdots$ | 0.724 | 0.970 |
| | Adv FGSM2 | 0.010 | 0.866 | 0.006 | $\cdots$ | 0.604 | 0.871 |
| | Adv FGSM80 | 0.008 | 0.653 | 0.559 | $\cdots$ | 0.023 | 0.089 |
| | LWA FGSM | 0.009 | 0.248 | 0.260 | $\cdots$ | 0.432 | 0.035 |
| | Sens FGSM | 0.009 | 0.266 | 0.285 | $\cdots$ | 0.365 | 0.039 |
| $\eta$=0.6 | No defense | 0.026 | 1.000 | 0.963 | $\cdots$ | 0.803 | 1.000 |
| | Adv FGSM1 | 0.012 | 0.003 | 0.889 | $\cdots$ | 0.790 | 0.889 |
| | Adv FGSM2 | 0.008 | 0.649 | 0.007 | $\cdots$ | 0.687 | 0.767 |
| | Adv FGSM80 | 0.009 | 0.439 | 0.426 | $\cdots$ | 0.020 | 0.021 |
| | LWA FGSM | 0.011 | 0.317 | 0.315 | $\cdots$ | 0.488 | 0.034 |
| | Sens FGSM | 0.010 | 0.264 | 0.244 | $\cdots$ | 0.465 | 0.033 |

Table 3: Error rates of different attacks on various adversarially-trained classifiers for MNIST. FGSM-curr means the FGSM attack on the specific classifier on the left. Adv FGSM is the classifier adversarially trained with FGSM attacks. Sens FGSM is the result of minimizing (7) by gradient descent (5). LWA FGSM is the result of minimizing (7) without the gradient-norm term.

## 4 GAME FORMULATION

In this section, we consider the class of optimization-based attack and the class of neural-network based attacks as an approximation of the former. Using the neural-network based attack class, we formulate the attacker-defender dynamics as a game and discuss two types of equilibria – the minimax and the maximin solutions. We present algorithms that generalize the approach presented in the previous section.

### 4.1 LEARNING-BASED ATTACK

An attacker $z(x) : \mathcal{X} \to \mathcal{X}$ can be more general than a specific class of attacks such as FGSM. Again, let $g : \mathcal{X} \to \mathcal{Y}$ is a classifier parameterized by $u$ and $l(g(x; u), y)$ is a loss function. If time complexity is not an issue, the following optimization-based attack (Huang et al., 2015)

$$\max_{Z=(z_1,...,z_N)} \left[ f(u, Z) \triangleq \frac{1}{N} \sum_i l(g(z_i; u), y_i) \right] = \frac{1}{N} \sum_i \max_{z_i} l(g(z_i; u), y_i), \qquad (8)$$

which is also related to the CW attack (Carlini & Wagner, 2017), can generate strong adversarial examples, where adversarial patterns $Z = (z_1, ..., z_N)$ are unrestricted except for the bounds such as $\|z_i - x_i\|_p \leq \eta$. The corresponding class of adversarial patterns $Z$ is very large, which results in strong but non-generalizable adversarial examples. Non-generalizable means the perturbation $z(x)$ has to be recomputed for every new test sample $x$. While the class of optimization-based attacks is powerful, its large size makes it difficult to analytically study the optimal defense methods. To make the problem *learnable*, we restrict the class of patterns $Z$ to that which can be generated by a flexible but manageable class of perturbation $\{z(\cdot; v) \mid \forall v \in V\}$, e.g., an autoencoder of a fixed architecture where the parameter $v$ is the network weights. This class is a clearly an approximation to the class of full optimization-based attacks, but is generalizable, i.e., no time-consuming optimization is required in the test phase but only single feedforward passes. The attack network (AttNet), as we

call it, can be of any class of appropriate neural networks. Here we use a three-layer fully-connected network with 300 hiddens units per layer in this paper. Different from Nguyen & Sinha (2017) or Baluja & Fischer (2017), we feed the label $y$ into the input of the network along with the features $x$. This is analogous to using the true label $y$ in the original FGSM. While this label input is optional but it can make the training of the attacker network easier. As with other attacks, we impose the $l_\infty$-norm constraint on $z$, i.e., $\|z(x) - x\|_\infty \leq \eta$.

Suppose now $f(u, v)$ is the empirical risk of a classifier-attacker pair where the input $x$ is first transformed by attack network $z(x; v)$ and then fed to the classifier $g(z(x; v); u)$. The attack network can be trained by gradient descent as well. Given a classifier $u$, we can use gradient descent

$$v \leftarrow v + \sigma \frac{\partial f(u, v)}{\partial v} \tag{9}$$

to find an optimal attacker $v$ that *maximizes* the risk $f$ assuming the classifier $u$ is fixed. Table 4 compares the error rates of the FGSM attacks and the attack network (AttNet). The table shows that AttNet is better than or comparable to FGSM in all cases. In particular, we already observed that the FGSM attack is no more effective against the classifier hardened against gradient-based attacks (Adv FGSM80 or Sens FGSM), but the AttNet can incur significant error ($> \sim 0.9$) for those hardened defenders. This indicates that the class of learning-based attacks is indeed different from the class of gradient-based attacks.

| Defense\Attack | FGSM-curr | AttNet-curr | FGSM-curr | AttNet-curr |
|---|---|---|---|---|
| | $\eta$=0.3 | | $\eta$=0.4 | |
| No defense | 1.000 | 1.000 | 1.000 | 1.000 |
| Adv FGSM1 | 0.996 | 1.000 | 0.984 | 1.000 |
| Adv FGSM80 | 0.473 | 0.899 | 0.131 | 0.903 |
| Sens FGSM | 0.048 | 0.965 | 0.038 | 0.902 |
| | $\eta$=0.5 | | $\eta$=0.6 | |
| No defense | 1.000 | 1.000 | 1.000 | 1.000 |
| Adv FGSM1 | 0.985 | 1.000 | 0.966 | 1.000 |
| Adv FGSM80 | 0.089 | 0.897 | 0.021 | 0.897 |
| Sens FGSM | 0.039 | 1.000 | 0.033 | 0.903 |

Table 4: Error rates of FGSM vs learning-based attack network (AttNet) on various adversarially-trained classifiers for MNIST. FGSM-curr/AttNet-curr means they are computed/trained for the specific classifier on the leftmost column. Note that FGSM fails to attack hardened networks (Adv FGSM80 and Sens FGSM), whereas AttNet can still attack them successfully.

## 4.2 MINIMAX GAME FOR LEARNING-BASED ATTACKS

Finally, we consider the dynamics of the pair of classifier-attacker when each player can change its parameters. Given the current classifier $u$, an optimal whitebox attacker parameter $v$ is the maximizer of the risk $f(u, v)$

$$v^*(u) \triangleq \arg\max_v f(u, v). \tag{10}$$

Consequently, the defender should choose the classifier parameters $u$ such that the maximum risk is minimized

$$u^* \triangleq \arg\min_u \max_v f(u, v) = \arg\min_u f(u, v^*(u)). \tag{11}$$

This solution to the continuous minimax problem has a natural interpretation as the best worst-case solution. Assuming the attacker is optimal, i.e., it chooses the best attack from (10) given $u$, no other defense can achieve a lower risk than the minimax defense $u^*$ in (11). The minimax defense is also a conservative defense. If the attacker is not optimal, and/or if the attack does not know the defense $u$ exactly (as in blackbox attacks), the actual risk can be lower than what the minimax solution $f(u^*, v^*(u^*))$ predicts. Before proceeding further, we point out that the claims above apply to the global minimizer $u^*$ and the maximizer function $v^*(\cdot)$, but in practice we can only find local solutions for complex risk functions of deep classifiers and attackers.

To solve (11), we analyze the problem similarly to (5)-(7) from the previous section. At each iteration, the defender should choose $u$ in expectation of the attack and minimize $f(u, v^*(u))$. We use

gradient descent

$$u \leftarrow u - \lambda \frac{df(u, v^*(u))}{du}, \tag{12}$$

where the total derivative $\frac{df}{du}$ is

$$\frac{df}{du} = \frac{\partial f(u, v^*(u))}{\partial u} + \frac{\partial v^*(u)}{\partial u} \frac{\partial f(u, v)}{\partial v}. \tag{13}$$

Since the exact maximizer $v^*(u)$ is difficult to find, we only update $v$ incrementally by one (or more) steps of gradient-ascent update

$$v \leftarrow v + \sigma \frac{\partial f(u, v)}{\partial v}. \tag{14}$$

The resulting formulation is closely related to the unrolled optimization (Metz et al., 2016) proposed for training GANs, although the latter has a very different cost function $f$. Using the single update (14), the total derivative is

$$\frac{df}{du} = \frac{\partial f(u, v^*(u))}{\partial u} + \sigma \frac{\partial^2 f(u, v)}{\partial u \partial v} \frac{\partial f(u, v)}{\partial v}. \tag{15}$$

Similar to hardening a classifier against gradient-based attacks by minimizing (7) at each iteration, the gradient update of $u$ for $f(u, v)$ can be done using the gradient of the following sensitivity-penalized function

$$f_{\text{sens}}(u) \triangleq f(u, v) + \frac{\sigma}{2} \left\| \frac{\partial f(u, v)}{\partial v} \right\|^2. \tag{16}$$

In other words, $u$ is chosen not only to minimize the risk but also to prevent the attacker from exploiting the sensitivity of $f$ to $v$. The algorithm is summarized in Alg. 1.

---

**Algorithm 1** Minimax Optimization by Sensitivity Penalization

---

Input: risk $f(u, v)$, # of iterations $T$, learning rates $(\sigma_i), (\lambda_i), (\gamma_i)$
Output: $(u^*, v^*(u^*))$
Initialize $u_0, v_0$
Begin
    **for** $i = 1, \ldots, T$ **do**
      *Max step:* $v_i = v_{i-1} + \sigma_i \frac{\partial f(u_{i-1}, v_{i-1})}{\partial v}$
      *Min step:* $u_i = u_{i-1} - \lambda_i \frac{\partial}{\partial u} \left[ f(u_{i-1}, v_{i-1}) + \frac{\gamma_{i-1}}{2} \left\| \frac{\partial f(u_{i-1}, v_{i-1})}{\partial v} \right\|^2 \right]$.
    **end for**
    Return $(u_T, v_T)$.

---

Note that this algorithm is actually independent of the adversarial example problem, and can be used for other minimax problems as well.

## 4.3 MINIMAX VS MAXIMIN PROBLEMS

In analogy with the minimax problem, we can also consider the maximin solution defined by

$$v^* \triangleq \arg\max_v \min_u f(u, v) = \arg\max_v f(u^*(v), v). \tag{17}$$

where

$$u^*(v) \triangleq \arg\min_u f(u, v) \tag{18}$$

is the minimizer function. Here we are abusing the notations for the minimax solution $u^*$, the maximin solution $v^*$, the minimizer $u^*(\cdot)$, and the maximizer $v^*(\cdot)$. Similar to the minimax solution, the maximin solution has an intuitive meaning – it is the best worst-case solution for the attacker. Assuming the defender is optimal, i.e., it chooses the best defense from (18) that minimizes the risk $f(u, v)$ given the attack $v$, no other attack can inflict a higher risk than the maximin attack $v^*$. It is also a conservative attack. If the defender is not optimal, and/or if the defender does not know the attack $v$ exactly, the actual risk can be higher than what the solution $f(u^*(v^*), v^*)$ predicts. Note

that the maximin scenario where the defender knows the attack method is not very realistic but is the opposite of the minimax scenario and provides the lower bound.

To summarize, minimax and maximin defenses and attacks have the following inherent properties.

**Lemma 1.** *Let $u^*, v^*(u), v^*, u^*(v)$ be the solutions of (11),(10),(17),(18).*

1. $f(u, v^*(u)) \geq f(u, v)$: *For any given defense $u$, the max attack $v^*(u)$ is the most effective attack.*

2. $f(u^*, v^*(u^*)) \leq f(u, v^*(u))$: *Against the optimal attack $v^*(u)$, the minimax defense $u^*$ is the most effective defense.*

3. $f(u^*(v), v) \leq f(u, v)$: *For any given attack $v$, the min defense $u^*(v)$ is the most effective defense.*

4. $f(u^*(v), v^*) \geq f(u^*(v), v)$: *Against the optimal defense $u^*(v)$, the maximin attack $v^*$ is the most effective attack.*

5. $\max_v \min_u f(u, v) \leq \min_u \max_v f(u, v)$: *The risk of the best worst-case attack is lower than that of the best worst-case defense.*

These properties follow directly from the definitions. The lemma helps us to better understand the dependence of defense and attack, and gives us the range of the possible risk values which can be measured empirically. To find maximin solutions, we use the same algorithm (Alg. 1) except that the variables $u$ and $v$ are switched and the sign of $f$ is flipped before the algorithm is called.

## 4.4 EXPERIMENTS

In addition to minimax and maximin optimization, we also consider as a reference algorithm the alternating descent/ascent method used in GAN training Goodfellow et al. (2014a)

$$u \leftarrow u - \lambda \frac{\partial f}{\partial u}, \quad v \leftarrow v + \sigma \frac{\partial f}{\partial v}. \tag{19}$$

Note that alternating descent/ascent finds local saddle points which are not necessarily minimax or maximin solutions, and therefore its solution will in general be different from the solution from Alg. 1. The difference of the solutions from three optimizations – Minimax, Maximin, and Alternating descent/ascent (Alt) – applied to a common problem, is demonstrated in Fig. 3. The figure shows the test error over the course of optimization starting from random initializations. One can see that Minimax (top blue curves) and Alt (middle green curves) converge to different values suggesting the learned classifiers will also be different.

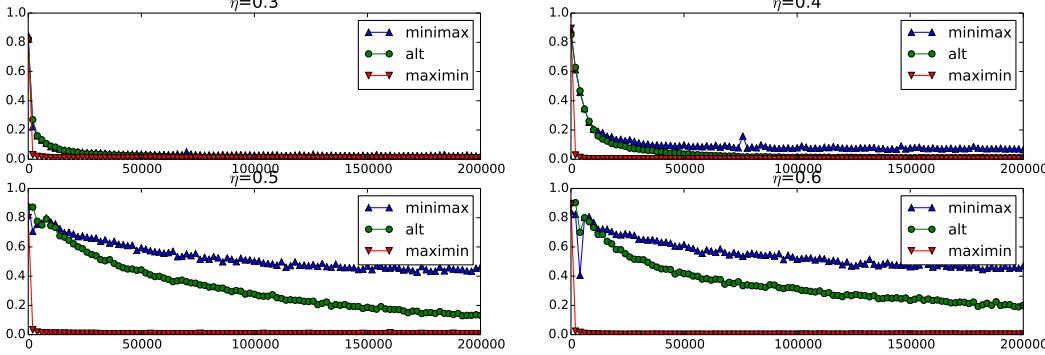

Figure 3: Convergence of the test error rates for Minimax optimization (blue), Alternating ascent/descent (green), and Maximin optimization (red) for MNIST.

Table 5 compares the robustness of the classifiers trained by Minimax and Alt against the AttNet attack (1st/2nd rows and 2nd column for each $\eta$.) Minimax defense is more robust than Alt defense

at $\eta = 0.3$ (0.020 vs 0.104) and at $\eta = 0.4$ (0.552 vs 0.873). For larger $\eta$'s, both are unusably vulnerable. Different performance of the two classifiers implies that the minimax solution found by Alg. 1 is different from the local saddle point found by alternating descent/ascent. In addition, against FGSM attacks, Minimax is moderately robust (0.218 – 0.342) despite that the classifiers are not specifically trained against gradient-based attacks. In contrast, Sens FGSM is very vulnerable (0.902 – 1.000) against AttNet which we have already observed. This result suggests that the class of AttNet attacks and the class of gradient-based attacks are indeed different, and the former class is larger than the latter.

| Defense\Attack | FGSM-curr | AttNet-curr | FGSM-curr | AttNet-curr |
|---|---|---|---|---|
| | $\eta$=0.3 | | $\eta$=0.4 | |
| Minimax | 0.218 | 0.020 | 0.238 | 0.552 |
| Alt | 0.244 | 0.104 | 0.503 | 0.873 |
| Sens FGSM | 0.048 | 0.965 | 0.038 | 0.902 |
| | $\eta$=0.5 | | $\eta$=0.6 | |
| Minimax | 0.342 | 1.000 | 0.299 | 1.000 |
| Alt | 0.289 | 0.902 | 0.157 | 0.899 |
| Sens FGSM | 0.039 | 1.000 | 0.033 | 0.903 |

Table 5: Error rates of Minimax-, Alt-, and adversarially-trained (Sens FGSM) classifiers for MNIST. Minimax is overall better than Alt against AttNet-curr, and is also moderately robust against the out-of-class attack (FGSM-curr).

Lastly, the adversarial examples generated by various attacks in the paper have diverse patterns and are shown in Fig. 4 of the appendix.

## 5 DISCUSSION

### 5.1 ROBUSTNESS AGAINST MULTIPLE ATTACK TYPES

We discuss some limitations of the framework and also propose an extension. Ideally, a defender should find a robust classifier against the worst attack from a very large class of attacks such as optimization-based attacks. However, it is difficult to train classifiers against attacks from a large class. On the other hand, if the class is too small, then the worst attack from that class is not representative of all possible worst attacks, and therefore the minimax defense found will not be robust to out-of-class attacks. The trade-off seems inevitable.

It is, however, possible to build a defense against multiple specific types of attacks. Suppose $z_1(u), ..., z_m(u)$ are $m$ different types of attacks, e.g., $z_1$=FGSM, $z_2$=IFGSM, etc. The minimax defense for the combined attack is the solution to the mixed continuous-discrete problem

$$\min_u \max\{f(u, z_1(u)), ..., f(u, z_m(u))\}. \tag{20}$$

Additionally, suppose $z_{m+1}(u, v), ..., z_{m+n}(u, v)$ are $n$ different types of learning-based attacks, e.g., $z_{m+1}$=2-layer dense net, $z_{m+2}$=5-layer convolutional nets, etc. The minimax defense against the mixture of multiple fixed-type and learning-based attacks can be found by solving

$$\min_u \max\{f(u, z_1(u)), ... , f(u, z_m(u)), \max_v f(u, z_{m+1}(u, v)), ... , \max_v f(u, z_{m+n}(u, v))\}. \tag{21}$$

Due to the huge computational demand to solve (21), we leave it as a future work.

### 5.2 ADVERSARIAL EXAMPLES AND PRIVACY ATTACKS

Lastly, we discuss a bigger picture of the game between adversarial players. The minimax optimization arises in the leader-follower game (Brückner & Scheffer, 2011) with the constant sum constraint. The leader-follower setting makes sense because the defense (=classifier parameters) is often public knowledge and the attacker exploits the knowledge. Interestingly, the problem of the attack on privacy (Hamm, 2016) has a very similar formulation as the adversarial attack problem, different only in that the classifier is an attacker and the data perturbator is a defender. In the problem of

privacy preservation against inference, the defender is a data transformer $z(x)$ (parameterized by $u$) which perturbs the raw data, and the attacker is a classifier (parameterized by $v$) who tries to extract sensitive information such as identity from the perturbed data such as online activity of a person. The transformer is the leader, such as when the privacy mechanism is public knowledge, and the classifier is the follower as it attacks the given perturbed data. The risk for the defender is therefore the accuracy of the inference of sensitive information measured by $-E[l(z(x;u),y;v)]$. Solving the minimax risk problem ($\min_u \max_v -E[l(z(x;u),y;v)]$) gives us the best worst-case defense when the classifier/attacker knows the transformer/defender parameters, which therefore gives us a robust data transformer to preserve the privacy against the best inference attack (among the given class of attacks.) On the other hand, solving the maximin risk problem ($\max_v \min_u -E[l(z(x;u),y;v)]$) gives us the best worst-case classifier/attacker when its parameters are known to the transformer. As one can see, the problems of adversarial attack and privacy attack are two sides of the same coin which can be addressed by similar frameworks and optimization algorithms.

## 6 CONCLUSION

In this paper, we present a continuous game formulation of adversarial attacks and defenses using a learning-based attack class implemented by neural networks. We show that this class of attacks is quite different from the gradient-based attacks. While a classifier robust to all types of attack may yet be an elusive goal, the minimax defense against the neural network-based attack class is well-defined and practically achievable. We show that the proposed optimization method can find minimax defenses which are more robust than adversarially-trained classifiers and the classifiers from simple alternating descent/ascent. We demonstrate these with MNIST and CIFAR-10.

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

# A    RESULTS WITH MNIST

The architecture of the MNIST classifier is similar to the Tensorflow model [2], and is trained with the following hyperparameters:
{*Batch size = 128, optimizer = AdamOptimizer with $\lambda = 10^{-4}$, total # of iterations=50,000.*}

The attack network has three hidden fully-connected layers of 300 units, trained with the following hyperparameters:
{*Batch size = 128, dropout rate = 0.5, optimizer = AdamOptimizer with $10^{-3}$, total # of iterations=30,000.*}

For minimax, alt, and maximin optimization, the total number of iteration was 100,000. The sensitivity-penalty coefficient of $\gamma = 1$ was used in Alg. 1.

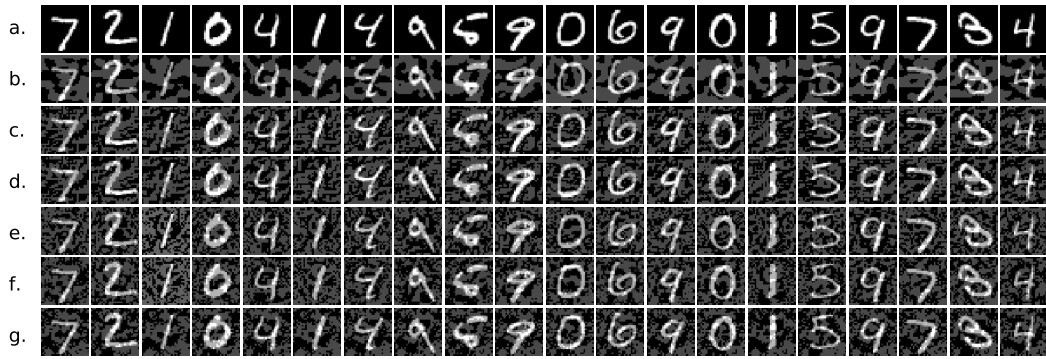

Figure 4: Adversarial samples generated from different attacks at $\eta = 0.2$. (a) Original data (b) FGSM1 (c) FGSM80 (d) IFGSM1 (e) Minimax (f) Alt (g) Maximin. Note the diversity of patterns.

# B    RESULTS WITH CIFAR-10

We preprocess the CIFAR-10 dataset by removing the mean and normalizing the pixel values with the standard deviation of all pixels in the image. It is followed by clipping the values to $\pm 2$ standard deviations and rescaling to $[-1, 1]$. The architecture of the CIFAR classifier is similar to the Tensorflow model [3] but is simplified further by removing the local response normalization layers. With the simple structure, we attained $\sim 78\%$ accuracy with the test data. The classifier is trained with the following hyperparameters:
{*Batch size = 128, optimizer = AdamOptimizer with $\lambda = 10^{-4}$, total # of iterations=100,000.*}

The attack network has three hidden fully-connected layers of 300 units, trained with the following hyperparameters:
{*Batch size = 128, dropout rate = 0.5, optimizer = AdamOptimizer with $\sigma = 10^{-3}$, total # of iterations=30,000.*}

For minimax, alt, and maximin optimization, the total number of iteration was 100,000. The sensitivity-penalty coefficient of $\gamma = 1$ was used in Alg. 1.

In the rest of the appendix, we repeat all the experiments with the MNIST dataset using the CIFAR-10 dataset.

---

[2]https://github.com/tensorflow/models/tree/master/tutorials/image/mnist
[3]https://github.com/tensorflow/models/tree/master/tutorials/image/cifar10

| Defense\Attack | No attack | FGSM | | | | IFGSM | | | |
|---|---|---|---|---|---|---|---|---|---|
| | | $\eta$=0.1 | $\eta$=0.2 | $\eta$=0.3 | $\eta$=0.4 | $\eta$=0.1 | $\eta$=0.2 | $\eta$=0.3 | $\eta$=0.4 |
| No defense | 0.222 | 0.976 | 0.825 | 0.869 | 0.884 | 0.668 | 0.907 | 0.959 | 0.971 |

Table 6: Error rates of FGSM and IFGSM attacks on the original classifier for cifar10. These attacks can cause large misclassification for the given range of $\eta$.

| Defense\Attack | No attack | FGSM | | | | IFGSM | | | |
|---|---|---|---|---|---|---|---|---|---|
| | | $\eta$=0.1 | $\eta$=0.2 | $\eta$=0.3 | $\eta$=0.4 | $\eta$=0.1 | $\eta$=0.2 | $\eta$=0.3 | $\eta$=0.4 |
| Adv train | n/a | 0.196 | 0.642 | 0.668 | 0.702 | 0.373 | 0.658 | 0.741 | 0.750 |

Table 7: Error rates of FGSM and IFGSM attacks on the adversarially-trained classifiers for CIFAR-10. This defense can significantly lower the errors from the attacks, although not as low as the MNIST problem.

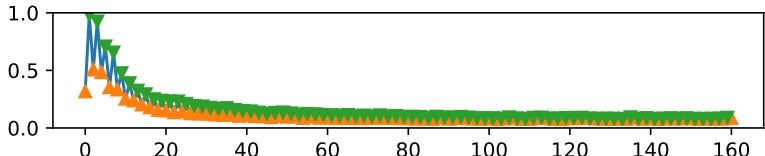

Figure 5: Cat and mouse game of FGSM attacks and adversarial training for CIFAR-10. The upper green points are the error rates after adversarial training, and the lower orange points are the error rates after FGSM attack. After 160 iterations ($\eta = 0.3$), the error rate is still oscillating.

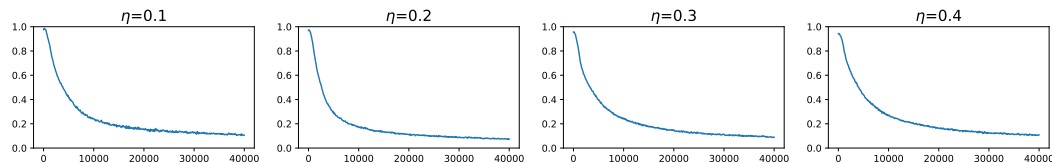

Figure 6: Convergence of test error rates for sensitivity-penalized optimization with MNIST.

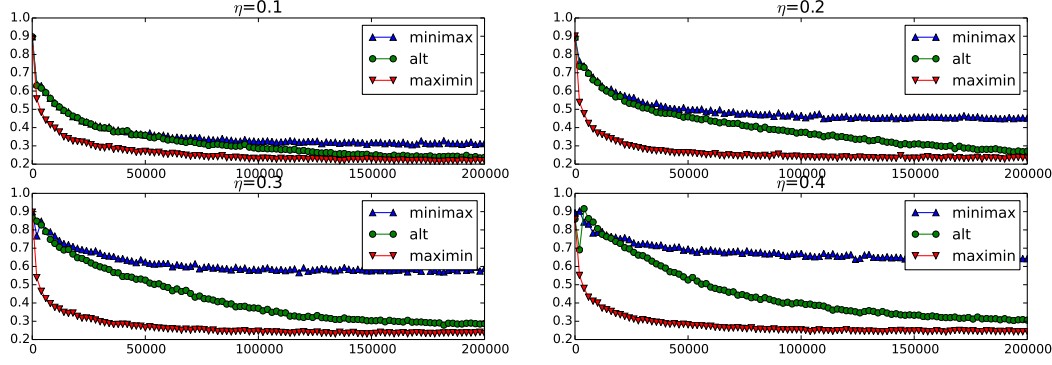

Figure 7: Convergence of the test error rates for Minimax optimization (blue), Alternating ascent/descent (green), and Maximin optimization (red) for CIFAR-10.

| Defense\Attack | No attack | FGSM | | | | FGSM-curr |
|---|---|---|---|---|---|---|
| | | FGSM-1 | FGSM-2 | ⋯ | FGSM-80 | |
| **η=0.1** No defense | 0.222 | 0.976 | 0.671 | ⋯ | 0.595 | 0.976 |
| Adv FGSM1 | 0.220 | 0.196 | 0.680 | ⋯ | 0.616 | 0.245 |
| Adv FGSM2 | 0.258 | 0.640 | 0.484 | ⋯ | 0.612 | 0.708 |
| Adv FGSM80 | 0.228 | 0.644 | 0.529 | ⋯ | 0.087 | 0.086 |
| LWA FGSM | 0.223 | 0.283 | 0.692 | ⋯ | 0.652 | 0.125 |
| Sens FGSM | 0.223 | 0.342 | 0.701 | ⋯ | 0.663 | 0.106 |
| **η=0.2** No defense | 0.222 | 0.825 | 0.692 | ⋯ | 0.819 | 0.969 |
| Adv FGSM1 | 0.216 | 0.642 | 0.630 | ⋯ | 0.609 | 0.264 |
| Adv FGSM2 | 0.305 | 0.579 | 0.290 | ⋯ | 0.599 | 0.556 |
| Adv FGSM80 | 0.218 | 0.445 | 0.502 | ⋯ | 0.078 | 0.078 |
| LWA FGSM | 0.209 | 0.689 | 0.666 | ⋯ | 0.615 | 0.105 |
| Sens FGSM | 0.209 | 0.713 | 0.672 | ⋯ | 0.637 | 0.073 |
| **η=0.3** No defense | 0.222 | 0.869 | 0.891 | ⋯ | 0.877 | 0.955 |
| Adv FGSM1 | 0.214 | 0.668 | 0.628 | ⋯ | 0.642 | 0.424 |
| Adv FGSM2 | 0.205 | 0.499 | 0.407 | ⋯ | 0.514 | 0.389 |
| Adv FGSM80 | 0.223 | 0.471 | 0.324 | ⋯ | 0.081 | 0.084 |
| LWA FGSM | 0.215 | 0.686 | 0.634 | ⋯ | 0.640 | 0.215 |
| Sens FGSM | 0.213 | 0.715 | 0.628 | ⋯ | 0.652 | 0.089 |
| **η=0.4** No defense | 0.222 | 0.884 | 0.899 | ⋯ | 0.892 | 0.941 |
| Adv FGSM1 | 0.208 | 0.702 | 0.687 | ⋯ | 0.697 | 0.536 |
| Adv FGSM2 | 0.206 | 0.592 | 0.546 | ⋯ | 0.618 | 0.545 |
| Adv FGSM80 | 0.225 | 0.497 | 0.385 | ⋯ | 0.121 | 0.124 |
| LWA FGSM | 0.210 | 0.693 | 0.639 | ⋯ | 0.626 | 0.173 |
| Sens FGSM | 0.214 | 0.714 | 0.635 | ⋯ | 0.640 | 0.109 |

Table 8: Error rates of different attacks on various adversarially-trained classifiers for CIFAR-10. FGSM-curr means the FGSM attack on the specific classifier on the leftmost column. Adv FGSM is the classifier adversarially trained with FGSM attacks. Sens FGSM is the result of minimizing the sensitivity penalty (7). LWA FGSM is the result of minimizing (7) without the gradient-norm term.

| Defense\Attack | FGSM-curr | AttNet-curr | FGSM-curr | AttNet-curr |
|---|---|---|---|---|
| | η=0.1 | | η=0.2 | |
| No defense | 0.976 | 0.740 | 0.969 | 0.905 |
| Adv FGSM1 | 0.245 | 0.999 | 0.264 | 1.000 |
| Adv FGSM80 | 0.086 | 1.000 | 0.078 | 1.000 |
| Sens FGSM | 0.106 | 0.898 | 0.073 | 0.979 |
| | η=0.3 | | η=0.4 | |
| No defense | 0.955 | 0.888 | 0.941 | 0.999 |
| Adv FGSM1 | 0.424 | 1.000 | 0.536 | 1.000 |
| Adv FGSM80 | 0.084 | 1.000 | 0.124 | 0.900 |
| Sens FGSM | 0.089 | 1.000 | 0.109 | 1.000 |

Table 9: Error rates of FGSM vs learning-based attack network (AttNet) on various adversarially-trained classifiers for CIFAR-10. FGSM-curr/AttNet-curr means they are computed/trained for the specific classifier on the leftmost column. Note that FGSM fails to attack against the 'hardened' networks (Adv FGSM80 and Sens FGSM), but AttNet can still attack them successfully.

| Defense\Attack | FGSM-curr | AttNet-curr | FGSM-curr | AttNet-curr |
|---|---|---|---|---|
| | η=0.1 | | η=0.2 | |
| Minimax | 0.967 | 0.276 | 0.980 | 0.418 |
| Alt | 0.994 | 0.264 | 0.996 | 0.857 |
| Sens FGSM | 0.106 | 0.898 | 0.073 | 0.979 |
| | η=0.3 | | η=0.4 | |
| Minimax | 0.967 | 0.875 | 0.931 | 0.994 |
| Alt | 0.987 | 0.896 | 0.958 | 1.000 |
| Sens FGSM | 0.089 | 1.000 | 0.109 | 1.000 |

Table 10: Error rates of Minimax-, Alt-, and adversarially-trained (Sens FGSM) classifiers for MNIST. While Minimax and Alt are both vulnerable to AttNet attacks, Minimax is much less vulnerable than Alt at $\eta = 0.2$.

