# OpenReview forum: "MACHINE VS MACHINE: MINIMAX-OPTIMAL DEFENSE AGAINST ADVERSARIAL EXAMPLES"
_ICLR.cc/2018/Conference — Reject_

### Official Review · AnonReviewer2 · 2017-11-26
**MACHINE VS MACHINE: DEFENDING CLASSIFIERS AGAINST LEARNING-BASED ADVERSARIAL ATTACKS**

**Rating:** 5
**Confidence:** 3

**Review:**

The game-theoretic approach to attacks with / defense against adversarial examples is an important direction of the security of deep learning and I appreciate the authors to initiate this kind of study.

Lemma 1 summarizes properties of the solutions that are expected to have after reaching equilibria. Important properties of saddle points in the min-max/max-min analysis assume that the function is convex/concave w.r.t. to the target variable.  In case of deep learning, the convexity is not guaranteed and the resulting solutions do not have necessarily follow Lemma 1.　Nonetheless, this type of analysis can be useful under appropriate solutions if non-trivial claims are derived; however, Lemma 1 simply explains basic properties of the min-max solutions and max-min solutions works and does not contain non-tibial claims.

As long as the analysis is experimental, the state of the art should be considered. As long as the reviewer knows, the CW attack gives the most powerful attack and this should be considered for comparison. The results with MNIST and CIFAR-10 are different. In some cases, MNIST is too easy to consider the complex structure of deep architectures. I prefer to have discussions on experimental results with both datasets.

The main takeaway from the entire paper is not clear very much. It contains a game-theoretic framework of adversarial examples/training, novel attack method, and many experimental results.

Minor:
Definition of g in the beginning of Sec 3.1 seems to be a typo. What is u? This is revealed in the latter sections but should be specified here.

In Section 3.1,
>This is in stark contrast with the near-perfect misclassification of the undefended classifier in Table 1.
The results shown in the table seems to indicate the “perfect” misclassification.

Sentence after eq. 15 seems to contain a grammatical error

The paragraph after eq. 17 is duplicated with a paragraph introduced before

---

> ### Author Response · Authors · 2017-12-13
> **The revision has clear conclusions that the initial submission was missing.**
>
> <Common>
>
> We thank all the reviewers for important suggestions.
> We could see where the submitted version was unclear or has caused confusions.
> Following the comments, we EXTENSIVELY revised the paper, re-ran the experiments and reported additional results to answer the questions.
> In particular, we show how the proposed minimax algorithm gives us better results than alternating descent/ascent used in GAN training, and how the class of neural-net based attacks is more general than the class of gradient-based attacks.
>
> Since we believe most of the questions are now addressed in the submitted revision, we politely ask the reviewers for updating their evaluations.
>
>
> <Reviewer 3>
>
> "Lemma 1 simply explains basic properties of the min-max solutions and max-min solutions works and does not contain non-tibial claims."
>
> Mathematically they are straightforward, although they have not been applied in this domain before.
> We agree this is not the essence of the experiments, and the new Table 5 now has more conclusive experimental results.
>
>
> "...  the CW attack gives the most powerful attack and this should be considered for comparison."
>
> AFAIK, there is no particularly effective method to optimization-based attacks [Huang'15, CW'15] when eta is large. In the revision, we discuss how the neural-network based attacks is an approximation of a much larger class of attacks such as CW, and how the approximation allows us to practically find minimax defenses. Direct adversarial training against optimization-based attacks [Huang'15] does not work, as shown in the new Table 3 (LWA FGSM).
>
> "The results with MNIST and CIFAR-10 are different. In some cases, MNIST is too easy to consider the complex structure of deep architectures. I prefer to have discussions on experimental results with both datasets."
>
> We understand but the paper is already 12 pages without the CIFAR-10 results. As one can see, our conclusions on the MNIST results are also applicable to the corresponding CIFAR-10 results, except that the error rates of different defenses/attacks are not as much spread as MNIST.
>
>
> "The main takeaway ... is not clear"
>
> We extensively revised the paper as well as reported important missing results. The key messages are 1) optimal defense-attack has to be studied as a dynamic problem, 2) we provide analytical and numerical tools to study them, and 3) the minimax defense is empirically better than previous adversarially-trained classifeirs or the results of optimization without sensitivity terms.
>
>
> "The paragraph after eq. 17 is duplicated with a paragraph introduced before"
>
> The paragraph (about maximin) is not the same as the previous paragraph (about minimax). They are exactly the opposite.

---

### Official Review · AnonReviewer1 · 2017-11-27
**Interesting extension and empirical study of GANs (Goodfellow et al. 2014)**

**Rating:** 6
**Confidence:** 3

**Review:**

This paper presents a sensitivity-penalized loss (the loss of the classifier has an additional term in squart of the gradient of the classifier w.r.t. perturbations of the inputs), and a minimax (or maximin) driven algorithm to find attacks and defenses. It has a lemma which claims that the "minimax and the maximin solutions provide the best worst-case defense and attack models, respectively", without proof, although that statement is supported experimentally.

+ Prior work seem adequately cited and compared to, but I am not really knowledgeable in the adversarial attacks subdomain.
- The experiments are on small/limited datasets (MNIST and CIFAR-10). Because of this, confidence intervals (over different initializations, for instance) would be a nice addition to Table 5.
- There is no exact ("alternating optimization" could be considered one) evaluation of the impact of the sensitivy loss vs. the minimax/maximin algorithm.
- The paper is hard to follow at times (and probably that dealing with the point above would help in this regard), e.g. Lemma 1 and experimental analysis.
- It is unclear (from Figures 3 and 7) that "alternative optimization" and "minimax" converged fully, and/or that the sets of hyperparameters were optimal.
+ This paper presents a game formulation of learning-based attacks and defense in the context of adversarial examples for neural networks, and empirical findings support its claims.


Nitpicks:
the gradient descent -> gradient descent or the gradient descent algorithm
seeming -> seemingly
arbitrary flexible -> arbitrarily flexible
can name "gradient descent that maximizes": gradient ascent.
The mini- max or the maximin solution is defined -> are defined
is the follow -> is the follower

---

> ### Author Response · Authors · 2017-12-13
> **Our minimax algorithm finds a more robust classifier than GAN-type alternating optimization**
>
> <Common>
>
> We thank all the reviewers for important suggestions.
> We could see where the submitted version was unclear or has caused confusions.
> Following the comments, we EXTENSIVELY revised the paper, re-ran the experiments and reported additional results to answer the questions.
> In particular, we show how the proposed minimax algorithm gives us better results than alternating descent/ascent used in GAN training, and how the class of neural-net based attacks is more general than the class of gradient-based attacks.
>
> Since we believe most of the questions are now addressed in the submitted revision, we politely ask the reviewers for updating their evaluations.
>
>
>
> <Reviewer 2>
>
> "The experiments are on small/limited datasets (MNIST and CIFAR-10). Because of this, confidence intervals (over different
> initializations, for instance) would be a nice addition to Table 5."
>
> We are in the process of repeating all the experiments and will report them as soon as they are available.
>
>
>
> "There is no exact evaluation of the impact of the sensitivity loss vs. the minimax/maximin algorithm."
>
> As for adversarial training against gradient-type attacks (Sec 3), the new Table 3 compares the classifiers trained with (Sens-FGSM) and without (LWA-FGSM) the sensitivity term, where the latter procedure is similar to Huang et al.'15. Sens-FGSM performs slightly better than LWA-FGSM.
> As for training against learning-based attacks (Sec 4), the new Table 5 compares the classifiers trained with (Minimax) and without (Alt) sensitivity term. The minimax solutions are shown to be more robust than the alt solutions. Fig 3 also shows that the solutions under the two methods converge to very different values.
>
>
> "...hard to follow ... Lemma 1 and experimental analysis."
>
> Lemma 1 follows simply from the definition and it was not the essence of the experiments. We replaced it with Table 5 which has more conclusive experimental results.
>
>
> "It is unclear (from Figures 3 and 7) that "alternative optimization" and "minimax" converged fully, and/or that the sets of hyperparameters were optimal."
>
> We tested the algorithms with different hyperparameters which did not improve the convergence speed. Instead, we now report the results with a 3-4 times larger number of iterations than before.

---

### Official Review · AnonReviewer3 · 2017-11-28
**Well-written, but experiments could be more thorough.**

**Rating:** 5
**Confidence:** 4

**Review:**

The authors describe a mechanism for defending against adversarial learning attacks on classifiers. They first consider the dynamics generated by the following procedure. They begin by training a classifier, generating attack samples using FGSM, then hardening the classifier by retraining with adversarial samples, generating new attack samples for the retrained classifier, and repeating.

They next observe that since FGSM is given by a simple perturbation of the sample point by the gradient of the loss, that the fixed point of the above dynamics can be optimized for directly using gradient descent. They call this approach Sens FGSM, and evaluate it empirically against the various iterates of the above approach.

They then generalize this approach to an arbitrary attacker strategy given by some parameter vector (e.g. a neural net for generating adversarial samples). In this case, the attacker and defender are playing a minimax game, and the authors propose finding the minimax (or maximin) parameters using an algorithm which alternates between maximization and minimization gradient steps. They conclude with empirical observations about the performance of this algorithm.

The paper is well-written and easy to follow. However, I found the empirical results to be a little underwhelming. Sens-FGSM outperforms the adversarial training defenses tuned for the “wrong” iteration, but it does not appear to perform particularly well with error rates well above 20%. How does it stack up against other defense approaches (e.g. https://arxiv.org/pdf/1705.09064.pdf)? Furthermore, what is the significance of FGSM-curr (FGSM-81) for Sens-FGSM? It is my understanding that Sens-FGSM is not trained to a particular iteration of the “cat-and-mouse” game. Why, then, does Sens-FGSM provide a consistently better defense against FGSM-81? With regards to the second part of the paper, using gradient methods to solve a minimax problem is not especially novel (i.e. Goodfellow et al.), thus I would liked to see more thorough experiments here as well. For example, it’s unlikely that the defender would ever know the attack network utilized by an attacker. How robust is the defense against samples generated by a different attack network? The authors seem to address this in section 5 by stating that the minimax solution is not meaningful for other network classes. However, this is a bit unsatisfying. Any defense can be *evaluated* against samples generated by any attacker strategy. Is it the case that the defenses fall flat against samples generated by different architectures?


Minor Comments:
Section 3.1, First Line. ”f(ul(g(x),y))” appears to be a mistake.

---

> ### Author Response · Authors · 2017-12-13
> **We added more experiments and revised the results.**
>
> <Common>
>
> We thank all the reviewers for important suggestions.
> We could see where the submitted version was unclear or has caused confusions.
> Following the comments, we EXTENSIVELY revised the paper, re-ran the experiments and reported additional results to answer the questions.
> In particular, we show how the proposed minimax algorithm gives us better results than alternating descent/ascent used in GAN training, and how the class of neural-net based attacks is more general than the class of gradient-based attacks.
>
> Since we believe most of the questions are now addressed in the submitted revision, we politely ask the reviewers for updating their evaluations.
>
>
> <Reviewer 1>
>
> "Sens-FGSM ... does not appear to perform particularly well with error rates well above 20%."
>
> Yes, that is true. With large eta's, hardening a classifier against all FGSM attacks by adversarial training is difficult, regardless of whether sensitivity norm is used or not.
>
>
> "https://arxiv.org/pdf/1705.09064.pdf"
>
> It is now included in the revision along with two other papers:
> "...A few researchers have also proposed using a detector to detect and reject adversarial examples \citep{meng2017magnet,lu2017safetynet,metzen2017detecting}. While we do not use detectors in this work, the minimax idea can be applied to train the detectors similarly."
>
>
> "Why ... does Sens-FGSM provide a consistently better defense aginst FGSM-81?"
>
> It's a misunderstanding. FGSM-curr for Sens-FGSM is the attack on the current parameter and not the same as FGSM-81.
> Anyway, Sens-FGSM is consistently better because it is trained so that the loss gradient is small.
>
>
> "... using gradient methods to solve a minimax problem is not especially novel (i.e. Goodfellow et al.)"
>
> It is not true. Alternating descent/ascent used in GAN cannot find minimax solutions but only local saddle points.
> Saddle points can be minimax, maximin or neither. They are the same only when f(u,v) is convex in u and concave in v.
> Empirically, the minimax and the alternating methods converge to very different values (Fig 3), and the minimax solutions are shown to be more robust than the alt solutions in the new Table 5.
>
>
> "it’s unlikely that the defender would ever know the attack network utilized by an attacker."
>
> Yes. The maximin case is the other extreme case which is more hypothetical than realistic. However it gives the lower bound.
>
>
> "How robust is the defense against samples generated by a different attack network?"
> "The authors ... state that the minimax solution is not meaningful for other network classes"
> "Is ... the defenses fall flat against samples generated by different architectures?"
>
> Sorry for the confusion. We unnecessarily overstated the limitations of minimax defense. They can certainly be evaluated against any other attack. We show in Table 5 that minimax-trained classifiers are still moderately robust to out-of-class FGSM attacks, whereas FGSM-trained classifiers fails utterly against neural-net based attacks. Evaluation with a different neural network architecture is underway.

---

### Public Comment · ~Seong_Joon_Oh1 · 2017-11-02
**untitled**

Thanks a lot for your great work! I think game theory is really one of the few valid ways to study attacks & defenses regarding adversarial examples, as opposed to the "cat-and-mouse game" we see in this field these days. Honestly, it is really becoming harder to trust papers saying "we have a great defense mechanism" or "we have a great attack method".

Along a similar line of reasoning, we have published a paper at ICCV'17, "Adversarial Image Perturbation for Privacy Protection -- A Game Theory Perspective". We have also proposed a game theoretic framework to find the equilibrium in the dynamics between user and recogniser, trying to thwart/re-enable recognition. Perhaps this paper should also be mentioned in the related work!

I'd like to point out some issues that I'd like to hear your response. First one is the term "best worst-case defense and attack". I feel this is contradictory to the fact that "we can only find local solutions in practice for complex loss functions such as deep networks-based defenders and attackers" (sec4.2). And this is also to me the biggest hurdle for using game theory with non-convex rewards under this security/privacy setup -- the equilibria, or the saddle points, do not guarantee anything, making the game theoretic analysis inconclusive.

Maybe a minor issue: while I like the cleanness of the formulation in eq7 ("sensitivity penality"), it eventually just tries to scale down the image gradients around the training data points (or hopefully around the entire data distribution). So, when FGSM is applied again to sensitivity penalised networks, wouldn't FGSM with larger step size (eta) re-enable high original error rate? Do you have any preliminary results?

While game theory has limitations (that it's hard to guarantee upper/lower bounds in non-convex setup), I still think game theory is great in spelling out assumptions explicitly (as we have argued in our ICCV'17 paper). I appreciate that the authors have really discussed the limitations in sec5.1. Overall, I really enjoyed the paper!

---

> ### Author Response · Authors · 2018-01-04
> **Game-theoretic view of adversarial examples**
>
> Thank you for your interest and comments. The ICCV'17 paper is certainly interesting and is in line with the main idea of the current paper that the adversarial example problems should be viewed as an attacker-defender game.
> The main difference between the ICCV'17 paper and the current paper is that we propose continuous minimax problems and new optimization algorithms as opposed to the classic discrete minimax problems on probability simplices used in the ICCV'17 paper. We will update our related work in the final version.
>
> Regarding local optima and the guarantee: We cannot at present expect to have the flexibility of non-convex models and the global optimality of convex-concave models at the same time. But in practice, we often observe that a local optimum of a neural network performs nearly as well as any other local optimum. It would be very desirable to have rigorous and tight bounds on the performance of deep neural networks.
>
> Regarding the formulation in eq7: We haven't tested it, but we believe that we can cause misclassification eventually by increasing eta. However, it also increases detectability of the presence of perturbation which defeats the purpose.
> A classifier with a lower sensitivity is certainly more robust to gradient-based attacks since it causes an attacker to use a stronger (eta) perturbation to fool the classifier.

---

> > ### Public Comment · ~Seong_Joon_Oh1 · 2018-01-05
> > **Remaining question**
> >
> > Thanks a lot for your kind explanations.
> >
> > I find it hard to agree with the last sentence though -- "A classifier with a lower sensitivity is certainly more robust to gradient-based attacks since it causes an attacker to use a stronger (eta) perturbation to fool the classifier. "
> >
> > To me, eta does not tell how strong the perturbation is -- the Lp norm of the perturbation does. If a network has been trained with the sensitivity penalty, then the image gradient itself will be downscaled. Then, even if a larger value of eta is applied, the Lp norm of the perturbation could still be smaller. My question is, does the sensitivity penalty improve the robustness of a network against attacks with the same Lp norm, rather than the same eta value (which is done in this work; please correct me if I'm wrong). If the robustness in this case remains the same, then I wouldn't say the sensitivity penalty works as a defence strategy.

---

> > > ### Author Response · Authors · 2018-01-05
> > > **eta vs Lp norm**
> > >
> > > Thanks again for the comment.
> > > We don't clearly understand the question since the Lp norm of the perturbation is always normalized to 1 in this paper.
> > > An adversarial sample z is z = x + eta*q, where x is the original image, eta is the perturbation strength, and q is the perturbation pattern with \|q\|_p=1. For FGSM, the pattern q = sign(grad loss) has a unit L-inf norm. After sensitivity training, the new q=sign(grad new loss) with a unit L-inf norm, cannot affect the new classifier as much as the old q (with a unit L-inf norm) could affect the undefended classifier using the same eta value. It would have to use a larger eta value.
> > > We hope this clarifies the issue.

---

> > > > ### Public Comment · ~Seong_Joon_Oh1 · 2018-01-05
> > > > **Makes sense now**
> > > >
> > > > That does clarify my misunderstanding. For a moment I thought the perturbations are scalar multiples of raw gradients, but yes it is FGSM indeed.
> > > > Thanks for your response!

---

### Decision · Program_Chairs · 2018-01-29
**ICLR 2018 Conference Acceptance Decision**

**Decision:**

Reject

**Comment:**

The paper studies a adversarial attacks and defenses against convolutional networks based on a minimax formulation of the problem. Whilst this is an interesting direction of research, the present paper seems preliminary. In particular, compared to several other independent ICLR submissions, the empirical evaluation is quite weak: it does not consider the strongest known gradient-based attack (Carlini-Wagner) as baseline and does not report results on ImageNet. The reviewers identify several issues related to Lemma 1 and to the clarity of presentation.